# GSPREC: TEMPORAL-AWARE GRAPH SPECTRAL FILTERING FOR RECOMMENDATION

## ABSTRACT

Graph-based recommendation systems are effective at modeling collaborative patterns but often suffer from two limitations: overreliance on low-pass filtering, which suppresses user-specific signals, and omission of sequential dynamics in graph construction. We introduce GSPRec, a graph spectral model that integrates temporal transitions through sequentially-informed graph construction and applies frequency-aware filtering in the spectral domain. GSPRec encodes item transitions via multi-hop diffusion to enable the use of symmetric Laplacians for spectral processing. To capture user preferences, we design a dual-filtering mechanism: a Gaussian bandpass filter to extract mid-frequency, user-level patterns, and a low-pass filter to retain global trends. Extensive experiments on four public datasets show that GSPRec consistently outperforms baselines, with an average improvement of 6.77% in NDCG@10. Ablation studies show the complementary benefits of both sequential graph augmentation and bandpass filtering.

## 1 INTRODUCTION

Graph-based recommender systems are constrained by two core problems: using low-pass spectral filtering often suppresses distinct user signals in favor of broad popularity patterns, and the order of user interactions is typically neglected in their graph structures. Therefore, recommender systems should leverage both collaborative patterns and the rich temporal signals embedded within these interaction sequences Koren (2009). Graph-based Collaborative Filtering (CF) methods He et al. (2020); Shen et al. (2021); Wang et al. (2019) have achieved strong performance, but they face two limitations. First, models such as LightGCN He et al. (2020), which rely on linear embedding propagation, act as low-pass filters Shen et al. (2021) by smoothing signals across the user-item graph. This encourages broad popularity effects (e.g., *everyone buys laptops*) but suppresses subgroup-specific patterns (e.g., *gamers buy laptops with mice*, *artists buy laptops with tablets*) Shen et al. (2021); Xia et al. (2025). Second, while some GCN-based recommenders incorporate temporal data, they do so via separate sequential modules Ma et al. (2020); Hsu & Li (2021) or through methods that do not integrate order into the graph structure for spectral analysis, which means they fail to preserve *sequential information* essential for distinguishing user preferences from popularity patterns.

The filtering behavior inherent in many GCNs motivates a closer look through the lens of graph signal processing (GSP), which generalizes classical signal processing to irregular graphs Ortega et al. (2018). A core technique in GSP is spectral filtering, which transforms graph signals using the eigenbasis of the graph Laplacian Kruzick & Moura (2017). While low-pass filters, widely adopted under the assumption of smoothness Shen et al. (2021); He et al. (2020), capture global popularity, they fail to retain the more distinctive user-specific variations Liu et al. (2023); Xia et al. (2025). Recent methods have explored amplifying high frequencies Guo et al. (2023), but risk magnifying noise. By contrast, mid-frequency components encode localized, task-relevant variations Dong et al. (2019) remains underexplored, motivating filters that explicitly target this band for personalization.

We introduce **GSPRec**, a novel GSP-based framework that achieves temporal awareness in spectral recommenders through two components: First, its *sequentially-aware graph construction* encodes multi-hop item transition patterns from user histories via diffusion into a symmetric graph structure to make temporal dynamics an intrinsic property for spectral analysis. Second, this graph empowers a *frequency-aware dual-filtering* by using Gaussian bandpass filter to extract mid-frequency user-specific preferences, complemented by a low-pass filter for popularity patterns.

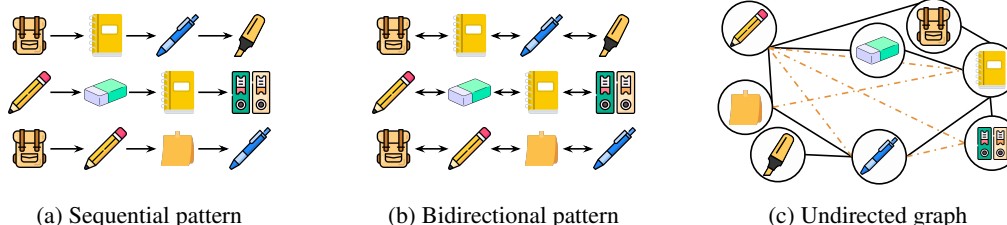

(a) Sequential pattern     (b) Bidirectional pattern     (c) Undirected graph

Figure 1: Conceptual shift from sequence-based to graph-based modeling. (a) Unidirectional transitions in RNNs Hidasi et al. (2015); Kang & McAuley (2018). (b) Bidirectional models like BERT4Rec Sun et al. (2019). (c) Our undirected item graph construction: solid lines show initial symmetric graph $\mathbf{S}'$, dashed lines show higher-order relationships from multi-hop diffusion.

In this work, we encode the interaction graph with item-to-item transitions derived from user interaction sequences. A multi-hop diffusion process with exponential decay captures sequential dependencies while preserving the benefits of symmetric graph structures. The symmetric structure guarantees real eigenvalues and orthogonal eigenvectors Chung (1997), which are important for stable spectral decomposition and well-defined filtering operations Ortega et al. (2018). Unlike prior work that extracts item-item similarity from collaborative information Xia et al. (2024), our approach aggregates explicit sequential signals into the graph topology[1]. Figure 1 illustrates this shift from sequence-based to graph-based modeling.

Moreover, we design a Gaussian bandpass filter tailored to recommendation graphs. Informed by empirical spectral energy distributions Qin et al. (2024); Tremblay et al. (2018), the filter selectively captures *mid-frequency components* that are often associated with user-specific and sequential patterns positioned between low-frequency popularity signals and high-frequency noise. The Gaussian profile provides optimal localization in both spectral and vertex domains Shuman et al. (2013); Ortega et al. (2018), which enables smooth transitions across frequencies and reduces ringing artifacts commonly introduced by sharp filter cutoffs Hammond et al. (2011). We summarize our contributions as follows:

- We propose a novel graph construction method that first derives a symmetric item connectivity graph from user sequences, then encodes higher-order sequential dependencies via multi-hop diffusion on this symmetric structure.

- We propose a Gaussian bandpass filter that targets mid-frequency components.

- We provide theoretical analysis establishing the stability and convergence of our sequential diffusion and filtering mechanisms.

- We validate our framework on four public benchmarks, showing that integrating sequential transitions with mid-frequency filtering leads to consistent performance gains over baselines.

GSPRec achieves average gains of 6.77% in NDCG@10 and 14.88% in MRR@10 across four public benchmark datasets, consistently outperforming state-of-the-art baselines. These results highlight the effectiveness of our two core design choices: sequential graph encoding and spectral filtering.

## 2 BACKGROUND & RELATED WORK

***Graph Signal Processing Background:*** Graph Signal Processing (GSP) analyzes signals on graph structures Ortega et al. (2018); Shuman et al. (2013). A *graph signal* maps values to nodes $\mathcal{G} = (\mathcal{V}, \mathcal{E})$ as $\mathbf{x} \in \mathbb{R}^{|\mathcal{V}|}$. The normalized Laplacian $\mathbf{L} = \mathbf{I} - \mathbf{D}^{-1/2}\mathbf{A}\mathbf{D}^{-1/2}$ decomposes into eigenvalues $\mathbf{\Lambda}$ and eigenvectors $\mathbf{U}$ Chung (1997), enabling spectral filtering:

$$\mathbf{x}' = \mathbf{U}\, g(\mathbf{\Lambda})\, \mathbf{U}^T \mathbf{x}, \quad \text{where } g(\mathbf{\Lambda}) = \text{diag}(g(\lambda_1), \ldots, g(\lambda_n)) \qquad (1)$$

Low-pass filters preserve global patterns, while band-pass filters capture mid-frequency variations related to user-specific behavior Ortega et al. (2018); Hammond et al. (2011).

---

[1]We note that sequences are used for graph construction, not for sequential next-item prediction.

Table 1: Comparison with existing popular methods.

| Method | Task | Spectral Filtering | Temporal Info. | Graph Structure | Message Passing | Mid-Freq |
|---|---|---|---|---|---|---|
| *Message Passing GCNs:* | | | | | | |
| NGCF Wang et al. (2019) | General | ✗ | ✗ | User-Item | ✓ | ✗ |
| LightGCN He et al. (2020) | General | ✗ | ✗ | User-Item | ✓ | ✗ |
| LR-GCCF Chen et al. (2020b) | General | ✗ | ✗ | User-Item | ✓ | ✗ |
| IMP-GCN Liu et al. (2021) | General | ✗ | ✗ | User-Item | ✓ | ✗ |
| *Spectral Filtering Methods (GSP-based):* | | | | | | |
| GF-CF Shen et al. (2021) | General | LP | ✗ | User-Item | ✗ | ✗ |
| JGCF Guo et al. (2023) | General | HP | ✗ | User-Item | ✗ | ✗ |
| PGSP Liu et al. (2023) | General | LP | ✗ | User-Item | ✗ | ✗ |
| HiGSP Xia et al. (2024) | General | LP | ✗ | User-Item | ✗ | ✗ |
| FaGSP Xia et al. (2025) | General | LP | ✗ | User-Item | ✗ | ✗ |
| *Sequential Recommendation:* | | | | | | |
| BERT4Rec Sun et al. (2019) | Sequential | ✗ | ✓ | Sequence | ✗ | ✗ |
| SASRec Kang & McAuley (2018) | Sequential | ✗ | ✓ | Sequence | ✗ | ✗ |
| **GSPRec (Ours)** | General | BP+LP | ✓ | UI+Seq | ✗ | ✓ |

***Graph-Based Collaborative Filtering:*** Graph-based collaborative filtering represents users and items as nodes in a bipartite graph, modeling interactions through various approaches He et al. (2020); Wang et al. (2019); Ying et al. (2018); Zheng et al. (2018). Recent works address over-smoothing Chen et al. (2020a); Shen et al. (2021), popularity bias Zhang et al. (2023), and efficiency Liang et al. (2024), while others incorporate temporal dynamics through sequential models or temporal graphs Kang & McAuley (2018); Sun et al. (2019); Zhang et al. (2024); Ou et al. (2025). Our approach encodes temporal locality as edge-level structure and applies spectral filtering without message passing.

***GSP in Recommendation:*** Prior work adapts GSP to recommender systems by treating interactions as graph signals. Most methods adopt low-pass filtering Huang et al. (2017); Liu et al. (2023); Xia et al. (2025; 2024), while recent work Guo et al. (2023) explores high-pass filtering, which may recover details but also amplify noise Shuman et al. (2013); Ortega et al. (2018). These approaches operate on static user-item graphs without sequential dynamics. In this work, we integrate sequential transitions into the graph before spectral analysis.

Table 1 summarizes existing approaches. Unlike GCN-based methods that rely on message passing, or GSP-based methods that apply low-pass (LP) or high-pass (HP) filtering to graphs, we join temporal graph construction (UI+Seq) with dual spectral filtering (BP+LP) to target mid-frequency components.

## 3 GSPREC

GSPRec consists of a sequential graph encoding procedure and a frequency-aware spectral filtering framework. These components jointly capture two complementary signal patterns in the spectral domain: (1) popularity patterns across the entire user-item graph, and (2) personalized patterns that distinguish individual preferences. Figure 2 summarizes the pipeline from user-item interactions to final recommendations via graph construction and spectral filtering.

### 3.1 PROBLEM FORMULATION

Let $\mathcal{U} = \{u_1, \ldots, u_m\}$ denote users and $\mathcal{I} = \{i_1, \ldots, i_n\}$ denote items. Historical interactions are triplets $\mathcal{D} = \{(u, i, t) \mid u \in \mathcal{U}, i \in \mathcal{I}, t \in \mathbb{R}^+\}$, yielding binary matrix $\mathbf{X} \in \{0, 1\}^{m \times n}$ where $x_{ui} = 1$ if user $u$ interacted with item $i$. For each user $u$, we define a time-ordered sequence $\mathcal{S}_u = [i_1, i_2, \ldots]$ where $(u, i_j, t_j) \in \mathcal{D}$ and $t_1 < t_2 < \cdots$. Our task is to predict future user-item interactions by jointly modeling collaborative patterns and sequential dynamics. We aim to predict unobserved entries in $\mathbf{X}$ using a unified spectral graph approach that leverages both user preference similarities and temporal item transitions.

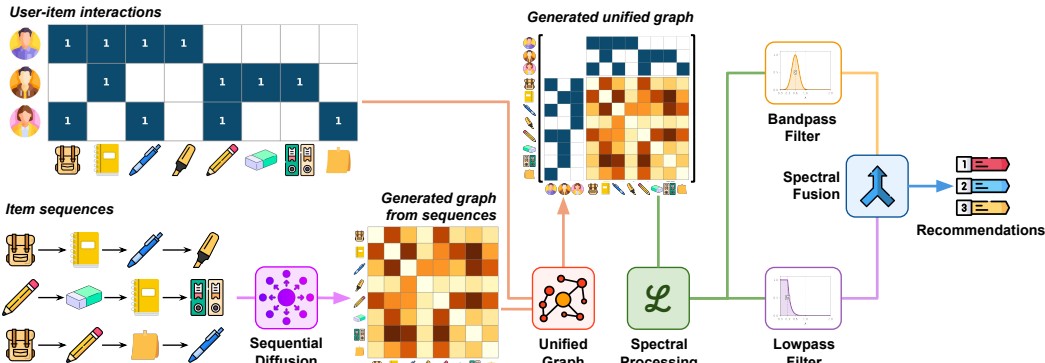

Figure 2: Overview of GSPRec. Starting from user-item interaction sequences (left), we construct a unified graph joining user-item links and item-item transitions via multi-hop diffusion with exponential decay (Eq. 2). This graph is processed using two complementary spectral filters: a band-pass filter to extract mid-frequency, user-specific patterns, and a low-pass filter to capture popularity.

## 3.2 RECOMMENDATION GRAPH CONSTRUCTION

Our goal is to construct a unified graph that integrates collaborative filtering signals with sequential transition patterns. Graph signal processing requires symmetric Laplacians to guarantee real eigenvalues, orthogonal eigenvectors, and stable spectral filtering Chung (1997); Ortega et al. (2018). Although user interaction histories are observed as timestamped sequences and thus appear directed, rigid order is not always reliable in recommendation domains Sun et al. (2019). We therefore transform sequences into a symmetric item-item graph component $\mathbf{S}$, a step that is theoretically required for spectral analysis. Multi-hop diffusion further preserves sequential proximity within this symmetric structure, allowing temporal information to remain embedded in the spectral representation.

Figure 1 illustrates our approach: while models in Figures 1a and 1b process sequences through uni/bidirectional mechanisms, we transform user interaction sequences into the undirected item graph in Figure 1c. This enables joint analysis of collaborative and sequential signals with well-defined spectral properties Kruzick & Moura (2017). Direct use of asymmetric Laplacians Chung (2005) for directed graphs involves different theoretical formulation left for future work.

**Initial Symmetric Sequence Graph ($\mathbf{S}'$):** We define an initial directed transition matrix $\mathbf{S} \in \{0, 1\}^{n \times n}$, where $s_{ij} = 1$ if item $i$ directly precedes item $j$ in any user sequence. To adapt this for symmetric spectral methods, we form its undirected version, $\mathbf{S}' \in \{0, 1\}^{n \times n}$. In $\mathbf{S}'$, an undirected edge $(i, j)$ exists if a transition $i \to j$ or $j \to i$ is in $\mathbf{S}$. This symmetrization is a standard practice when preparing directed graph for algorithms requiring undirected structures Von Luxburg (2007); Satuluri & Parthasarathy (2011). While simplifying transition directionality, this approach allows capturing broader item relatedness from sequences, as rigid order is not always optimal and wider context can be beneficial Sun et al. (2019). Multi-hop diffusion (Eq. 2) operates on $\mathbf{S}'$ to encode sequence-derived item associations.

For example, consider a user sequence backpack→notebook→pen illustrated in the running example in Figures 1–3. Symmetrization creates undirected edges (backpack, notebook) and (notebook, pen). The second-order diffusion step then adds a weighted edge (backpack, pen), which reflects the second-order temporal relationship derived from the original sequence. While this edge is undirected, its strength encodes the sequential context, allowing our subsequent spectral filters to leverage these embedded temporal patterns for recommendation.

**Multi-Hop Diffusion ($\mathbf{S}^{(d)}$):** To model broader item influence through multi-step paths, we apply multi-hop diffusion to $\mathbf{S}'$:

$$\mathbf{S}^{(d)} = \sum_{k=1}^{d} \alpha^{k-1}(\mathbf{S}')^{k} \tag{2}$$

where $\alpha \in (0, 1)$ is an exponential decay factor that gives higher weight to shorter paths in $\mathbf{S}'$, thus emphasizing local sequential relationships. $d$ is the diffusion depth to control the extent of influence propagation. Since $\mathbf{S}'$ is symmetric, each power $(\mathbf{S}')^k$ is symmetric, and therefore the diffused matrix $\mathbf{S}^{(d)}$ is also symmetric. The entries $S_{ij}^{(d)}$ represent a score of multi-hop, locality-aware proximity between items $i$ and $j$, learned from the aggregated sequential patterns as illustrated in Figure 3. This diffusion process is analogous to using graph diffusion kernels to define similarity Gasteiger et al. (2019) and converges when $\alpha < 1/\rho(\mathbf{S}')$. In practice, we estimate $\rho(\mathbf{S}')$ (e.g., via power iteration) and choose a suitable $\alpha$.

**Lemma 1** (Diffusion Convergence)*. The diffusion process defined in Eq. 2 converges as $d \to \infty$ if $\alpha < 1/\rho(\mathbf{S}')$, where $\rho(\cdot)$ denotes the spectral radius of $\mathbf{S}'$.*

*Proof.* The diffusion process as $d \to \infty$ is $\mathbf{S}^{(\infty)} = \sum_{k=1}^{\infty} \alpha^{k-1} (\mathbf{S}')^k$. We can rewrite this as $\mathbf{S}^{(\infty)} = \mathbf{S}' \sum_{j=0}^{\infty} (\alpha \mathbf{S}')^j$ by letting $j = k - 1$. This series converges if the standard matrix geometric series $\sum_{j=0}^{\infty} M^j$, where $M = \alpha \mathbf{S}'$, converges. This occurs if the spectral radius $\rho(M) < 1$, i.e., $\rho(\alpha \mathbf{S}') < 1$. Since $\rho(\alpha \mathbf{S}') = \alpha \rho(\mathbf{S}')$ (for $\alpha > 0$), convergence requires $\alpha < 1/\rho(\mathbf{S}')$. $\square$

**Symmetric Normalization ($\tilde{\mathbf{S}}$):** The resulting symmetric, diffused matrix $\mathbf{S}^{(d)}$ is then symmetrically normalized to produce the final item-item graph component:

$$\tilde{\mathbf{S}} = \mathbf{D}_S^{-1/2} \mathbf{S}^{(d)} \mathbf{D}_S^{-1/2}, \quad \text{where } \mathbf{D}_S = \text{diag}(\mathbf{S}^{(d)} \mathbf{1}_n) \tag{3}$$

This standard normalization technique Chung (1997) yields $\tilde{\mathbf{S}}$, ensuring its weights are scaled appropriately for stable spectral analysis. The weights in $\tilde{\mathbf{S}}$ reflect a transformation of user sequences into measures of item-item sequential relevance.

**Interaction Normalization:** The raw user-item interaction matrix $\mathbf{X}$ is normalized to account for varying user activity levels and popularities of the items. Standard normalizations include:

$$\tilde{\mathbf{X}}_U = \mathbf{D}_U^{-1/2} \mathbf{X}, \quad \tilde{\mathbf{X}}_I = \mathbf{X} \mathbf{D}_I^{-1/2} \tag{4}$$

where $\mathbf{D}_U = \text{diag}(\mathbf{X} \mathbf{1}_n)$ and $\mathbf{D}_I = \text{diag}(\mathbf{X}^T \mathbf{1}_m)$. For constructing the unified adjacency matrix $\mathbf{A}$, we use the original binary interaction matrix $\mathbf{X}$, as its structure is fundamental. The normalized versions can be used in other components, like the filter design.

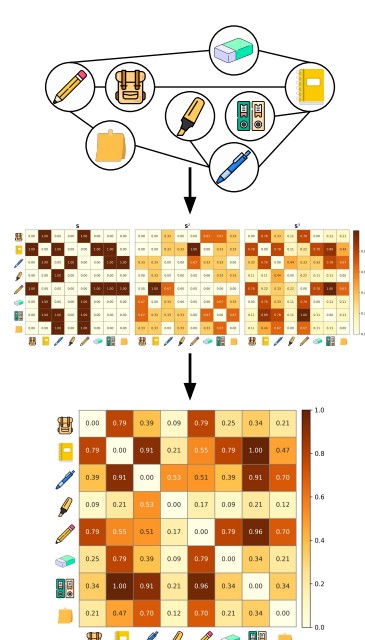

Figure 3: Multi-hop diffusion encoding item proximity from sequential patterns. Top: $\mathbf{S}'$. Middle: Powers $(\mathbf{S}')^k$. Bottom: $\mathbf{S}^{(3)}$ encoding decayed proximity.

**Unified Graph Adjacency and Laplacian ($\mathbf{A}, \mathbf{L}$):** We integrate the user-item interactions $\mathbf{X}$ and the sequence-derived item-item graph $\tilde{\mathbf{S}}$ into a single, symmetric adjacency matrix for the graph:

$$\mathbf{A} = \begin{bmatrix} \mathbf{0}_{m \times m} & \mathbf{X} \\ \mathbf{X}^T & \tilde{\mathbf{S}} \end{bmatrix} \tag{5}$$

This matrix $\mathbf{A}$ forms a heterogeneous unified graph where the $\mathbf{0}_{m \times m}$ block denotes no direct user-user links are explicitly modeled here, $\mathbf{X}$ captures user-item interactions, and $\tilde{\mathbf{S}}$ encodes the sequence-derived item-item relationships. From $\mathbf{A}$, we compute the symmetric normalized Laplacian, which is the cornerstone of our spectral filtering:

$$\mathbf{L} = \mathbf{I} - \mathbf{D}^{-1/2} \mathbf{A} \mathbf{D}^{-1/2}, \quad \text{where } \mathbf{D} = \text{diag}(\mathbf{A} \mathbf{1}) \tag{6}$$

The properties of this Laplacian are discussed in Proposition 2. Our approach, by constructing $\tilde{\mathbf{S}}$ from symmetrized sequential data and then applying diffusion, embeds sequence-derived proximity into a standard spectral framework, offering a distinct way to leverage temporal signals compared to pure sequential models that maintain strict causality Kang & McAuley (2018); Sun et al. (2019).

**Proposition 2** (Laplacian Validity). *The Laplacian* $\mathbf{L}$*, constructed from the symmetric adjacency matrix* $\mathbf{A}$*, is symmetric, positive semidefinite, and has real eigenvalues in the range* $[0, 2]$*.*

*Proof sketch; full proof in Appendix B.* This follows from spectral graph theory Chung (1997), as $\mathbf{A}$ is constructed to be symmetric and non-negative, and Eq. 6 defines the standard normalized Laplacian. $\square$

### 3.3 SPECTRAL FILTERING FRAMEWORK

To model user preferences at different levels of specificity, we perform spectral filtering on the Laplacian graph $\mathbf{L} = \mathbf{U}\mathbf{\Lambda}\mathbf{U}^T$ where $\mathbf{\Lambda} = \text{diag}(\lambda_1, \ldots, \lambda_{m+n})$ contains eigenvalues in ascending order $0 \leq \lambda_1 \leq \lambda_2 \leq \ldots \leq \lambda_{m+n} \leq 2$ and $\mathbf{U} = [\mathbf{u}_1, \ldots, \mathbf{u}_{m+n}]$ contains the corresponding eigenvectors. In practice, for computational efficiency, we perform a truncated eigendecomposition focusing on $r \ll (m+n)$ which includes the low- and mid-frequency bands relevant to our filters. The lowpass filter $g_{\text{LP}}(\lambda)$ and bandpass filter $g_{\text{BP}}(\lambda)$ operate on the corresponding $r$ eigenpairs by selecting and amplifying the mid-frequency signals present within this subspace.

**Proposition 3** (Laplacian Properties). *The eigenvalues of* $\mathbf{L}$ *lie in* $[0, 2]$*, with the number of zero eigenvalues equal to the number of connected components in the graph.*

**Bandpass Filtering:** To capture mid-frequency user-specific behavior, we design a bandpass filter using a Gaussian kernel:

$$g_{\text{BP}}(\lambda) = \exp\left(-\frac{(\bar{\lambda} - c)^2}{w}\right), \quad \bar{\lambda} = \frac{\lambda - \lambda_{\min}}{\lambda_{\max} - \lambda_{\min}} \tag{7}$$

where $c$ and $w$ control the center and width of the frequency band. Our empirical analysis (Appendix G) shows spectral coefficients for intermediate eigenvalues have higher magnitude for user-specific preferences than global trends. This aligns with spectral theory: low frequencies capture global popularity (smooth across graph), high frequencies represent noise (rapid variation), while mid-frequencies are piecewise smooth, *which varies between communities but consistent within them.* In recommendations, user communities correspond to groups with shared item preferences, making mid-frequency the natural band for personalization.

Let $\mathbf{G}_{\text{BP}} = \text{diag}(g_{\text{BP}}(\lambda_1), \ldots, g_{\text{BP}}(\lambda_r))$ denote the spectral bandpass filter. We filter the item-degree-normalized user-item interaction signal:

$$\mathbf{F}_{BP} = \mathbf{X}\mathbf{D}_I^{-1/2}\mathbf{U}\mathbf{G}_{BP}\mathbf{U}^T\mathbf{D}_I^{-1/2} \tag{8}$$

**Lowpass Filtering:** To preserve global structure and smooth preferences, we compute a complementary low-pass component augmented with user-user similarity. To enhance global structure, we augment user-item signals with user-user similarities. We define $\mathbf{X}_b = [\mathbf{C}_U, \mathbf{X}]$, where $\mathbf{C}_U = \tilde{\mathbf{X}}_U\tilde{\mathbf{X}}_U^T$. The low-pass component is then computed as:

$$\mathbf{F}_{LP} = \mathbf{X}_b\mathbf{D}_b^{-1/2}\mathbf{U}\mathbf{U}^T\mathbf{D}_b^{1/2} \tag{9}$$

where $\mathbf{D}_b = \text{diag}(\mathbf{X}_b^T\mathbf{1}_m)$. We note that $\mathbf{C}_U$ is not part of the Laplacian but is introduced to incorporate global user similarity in the lowpass representation. As $\mathbf{X}_b$ contains both user-user and user-item components, we extract only the item-related part after filtering by:

$$\mathbf{F}_{LP} = \mathbf{F}_{LP}[:, m :] \tag{10}$$

In our framework, mid-frequency components correspond to more personalized patterns, while low frequencies capture smoother, population-level preferences. We compute item scores by fusing mid-frequency personalization with low-frequency global trends. Specifically, we compute:

$$\mathbf{Y} = \phi \cdot \mathbf{F}_{\text{BP}} + (1 - \phi) \cdot \mathbf{F}_{\text{LP}}, \quad \phi \in [0, 1] \tag{11}$$

Both $\mathbf{F}_{\text{BP}}$ and $\mathbf{F}_{\text{LP}}$ are outputs of linear spectral filters applied to the graph Laplacian $\mathbf{L}$. Their convex combination remains a valid linear operator, as the space of spectral filters is closed under *addition* and *scalar multiplication* Sandryhaila & Moura (2013); Püschel & Moura (2006). Figure 2 illustrates our dual-filter architecture.

Table 2: Statistics of evaluation datasets.

| Dataset | #Users | #Items | #Inter. | Density |
|---|---|---|---|---|
| ML100K | 943 | 1,682 | 100,000 | 6.30% |
| ML1M | 6,040 | 3,706 | 1,000,209 | 4.47% |
| Netflix | 20,000 | 17,720 | 5,678,654 | 1.60% |
| Beauty | 22,363 | 12,101 | 198,502 | 0.07% |

Table 3: Filter parameters across datasets.

| Dataset | $\phi$ | $c$ | $w$ | $r$ |
|---|---|---|---|---|
| ML100K | 0.5 | 0.2 | 0.1 | 32 |
| ML1M | 0.3 | 0.8 | 0.1 | 128 |
| Netflix | 0.3 | 0.4 | 0.3 | 256 |
| Beauty | 0.5 | 0.8 | 0.3 | 512 |

The spectral components separate distinct preference signals: low-frequency components (from lowpass filter) capture common transitions shared across users like the backpack→notebook association from our running example. Mid-frequency components (from bandpass filter) extract distinctive patterns such as pencil→eraser→notebook→calculator sequences that characterize specific user subgroups. Figure 3 shows how our diffusion process captures these sequential patterns, with darker cells indicating stronger connections derived from user histories. Appendix B provides a complete description, with complexity analysis in Appendix D.

## 4 EXPERIMENTS

We evaluate GSPRec to answer: (1) How does sequential graph enrichment affect performance? (2) What is the impact of frequency-aware filtering? (3) How does GSPRec compare to state-of-the-art recommenders?

### 4.1 EXPERIMENTAL SETUP

**Data:** We conduct experiments on four benchmark datasets: ML100K, ML1M Harper & Konstan (2015), Netflix Bennett & Lanning (2007), and Amazon Beauty Ni et al. (2019). These datasets maintain comparability with recent spectral methods Xia et al. (2025; 2024). We use an 8:1:1 train/test split with 10% of training data for validation. Statistics are shown in Table 2.

**Baselines:** We compare GSPRec with: (i) **Popularity**; (ii) GCN-based methods: **LightGCN** He et al. (2020), **LR-GCCF** Chen et al. (2020b), **IMP-GCN** Liu et al. (2021), **SimpleX** Mao et al. (2021a), **UltraGCN** Mao et al. (2021b); and (iii) GSP-based methods: **GF-CF** Shen et al. (2021), **JGCF** Guo et al. (2023), **PGSP** Liu et al. (2023), **HiGSP** Xia et al. (2024), **FaGSP** Xia et al. (2025). Detailed descriptions are in Appendix E. GSPRec leverages temporal graph construction with dual-filter approach targeting both low- and band-pass components.

**Implementation Details:** For graph construction, we set the diffusion depth $d = 2$ and the decay $\alpha = 0.4$ in all datasets to capture immediate and second-order transitions while attenuating longer-range influences. Eigendecomposition dimensionality $r$ varies by dataset size. Filter parameters $c$, $w$, and fusion weight $\phi$ are tuned per dataset as shown in Table 3. Implementation details are in Appendix F.

**Metrics:** Following prior work He et al. (2020); Shen et al. (2021); Xia et al. (2025); Guo et al. (2023); Xia et al. (2024); Liu et al. (2023), we use Normalized Discounted Cumulative Gain (NDCG@$k$) and Mean Reciprocal Rank (MRR@$k$) at $k \in \{5, 10, 20\}$, abbreviated as N@k and M@k respectively. We apply spectral filtering to obtain preference scores, exclude previously interacted items, and rank the top-$k$ items.

### 4.2 OVERALL PERFORMANCE

Table 4 presents performance comparisons. GSPRec consistently outperforms all baselines across metrics and datasets, with improvements over recent GSP-based methods validating the effectiveness of leveraging sequential graph construction with mid-frequency spectral filtering.

Performance improvements range from 0.54% to 26.37% across datasets and metrics. Beauty dataset achieves 19.74% improvement on N@20, while ML100K shows improvements ranging from 5.26% to 26.37% and Netflix demonstrates gains of 0.54% to 16.30%. This variation indicates that method effectiveness depends on dataset-specific characteristics that extend beyond simple density metrics.

Table 4: Performance comparison on four public datasets. The best performance is denoted in **bold**. "Improv." denotes the percentage improvement of GSPRec compared to the strongest baseline method.

| Datasets | Metric | GCN-based Methods | | | | | | GSP-based Methods | | | | | | Improv. |
|---|---|---|---|---|---|---|---|---|---|---|---|---|---|---|
| | | Popularity | LightGCN | LR-GCCF | IMP-GCN | SimpleX | UltraGCN | GF-CF | JGCF | PGSP | FaGSP | HiGSP | GSPRec | |
| ML100K | N@5 | 0.5672 | 0.7034 | 0.5587 | 0.6764 | 0.6995 | 0.6786 | 0.6875 | 0.6582 | 0.6851 | 0.7106 | _0.7166_ | **0.7543** | +5.26% |
| | M@5 | 0.5151 | 0.6362 | 0.4943 | 0.6006 | 0.6397 | 0.6094 | 0.6234 | 0.4471 | 0.6149 | 0.6518 | _0.6629_ | **0.7358** | +11.00% |
| | N@10 | 0.5857 | 0.6771 | 0.5603 | 0.6605 | 0.6866 | 0.6688 | 0.6843 | 0.6695 | 0.6722 | _0.7092_ | 0.7021 | **0.7572** | +6.77% |
| | M@10 | 0.5296 | 0.5877 | 0.4616 | 0.5690 | 0.6064 | 0.5749 | 0.6010 | 0.4222 | 0.5767 | _0.6384_ | 0.6380 | **0.7443** | +16.59% |
| | N@20 | 0.5867 | 0.6618 | 0.5555 | 0.6605 | 0.6543 | 0.6486 | 0.6621 | 0.6730 | 0.6495 | _0.6989_ | 0.6724 | **0.7465** | +6.81% |
| | M@20 | 0.5315 | 0.5652 | 0.4411 | 0.5690 | 0.5411 | 0.5289 | 0.5593 | 0.4197 | 0.5496 | _0.5901_ | 0.5782 | **0.7457** | +26.37% |
| ML1M | N@5 | 0.0958 | 0.5845 | 0.3540 | 0.5583 | 0.5934 | 0.5739 | 0.5935 | 0.6121 | 0.5963 | _0.6112_ | 0.6062 | **0.6237** | +2.05% |
| | M@5 | 0.1943 | 0.5160 | 0.2997 | 0.4934 | 0.5244 | 0.5087 | 0.5254 | 0.4172 | 0.5313 | _0.5430_ | 0.5386 | **0.5715** | +5.25% |
| | N@10 | 0.0960 | 0.5873 | 0.3787 | 0.5594 | 0.5921 | 0.5773 | 0.5897 | _0.6238_ | 0.5923 | 0.6082 | 0.6042 | **0.6431** | +3.10% |
| | M@10 | 0.2180 | 0.5010 | 0.2946 | 0.4685 | 0.5051 | 0.4886 | 0.4996 | 0.3869 | 0.5063 | _0.5229_ | 0.5148 | **0.5837** | +11.63% |
| | N@20 | 0.1035 | 0.5692 | 0.3980 | 0.5416 | 0.5687 | 0.5600 | 0.5678 | _0.6270_ | 0.5687 | 0.5808 | 0.5769 | **0.6431** | +2.57% |
| | M@20 | 0.2309 | 0.4644 | 0.2924 | 0.4292 | 0.4625 | 0.4500 | 0.4557 | 0.3730 | 0.4646 | _0.4748_ | 0.4620 | **0.5871** | +23.65% |
| Netflix | N@5 | 0.5105 | 0.6822 | 0.6023 | 0.6328 | 0.6603 | 0.5680 | 0.7005 | 0.7065 | 0.6811 | _0.7162_ | 0.7162 | **0.7201** | +0.54% |
| | M@5 | 0.4505 | 0.6147 | 0.5323 | 0.5528 | 0.5902 | 0.4941 | 0.6360 | 0.5283 | 0.6127 | 0.6524 | _0.6532_ | **0.6680** | +2.27% |
| | N@10 | 0.5488 | 0.6814 | 0.6134 | 0.6307 | 0.6694 | 0.5746 | 0.6928 | _0.7148_ | 0.6756 | 0.7079 | 0.7062 | **0.7270** | +1.71% |
| | M@10 | 0.4712 | 0.5974 | 0.5234 | 0.5386 | 0.5839 | 0.4708 | 0.6126 | 0.4837 | 0.5905 | _0.6293_ | 0.6275 | **0.6762** | +7.45% |
| | N@20 | 0.5604 | 0.6642 | 0.6043 | 0.6152 | 0.6572 | 0.5580 | 0.6689 | _0.7164_ | 0.6543 | 0.6814 | 0.6800 | **0.7213** | +0.68% |
| | M@20 | 0.4773 | 0.5636 | 0.4963 | 0.4995 | 0.5555 | 0.4309 | 0.5691 | 0.4273 | 0.5502 | _0.5827_ | 0.5800 | **0.6777** | +16.30% |
| Beauty | N@5 | 0.0149 | 0.0668 | 0.0533 | 0.0570 | 0.0682 | 0.0599 | 0.0659 | 0.0501 | 0.0674 | 0.0712 | _0.0718_ | **0.0733** | +2.09% |
| | M@5 | 0.0118 | 0.0530 | 0.0423 | 0.0444 | 0.0544 | 0.0476 | 0.0523 | 0.0535 | 0.0564 | _0.0574_ | | **0.0619** | +7.84% |
| | N@10 | 0.0200 | 0.0763 | 0.0635 | 0.0667 | 0.0761 | 0.0649 | 0.0751 | 0.0612 | 0.0765 | _0.0799_ | 0.0771 | **0.0879** | +10.01% |
| | M@10 | 0.0139 | 0.0519 | 0.0443 | 0.0451 | 0.0517 | 0.0441 | 0.0517 | _0.0590_ | 0.0525 | 0.0549 | 0.0531 | **0.0680** | +15.25% |
| | N@20 | 0.0256 | 0.0829 | 0.0678 | 0.0729 | 0.0819 | 0.0689 | 0.0819 | 0.0721 | 0.0828 | _0.0846_ | 0.0799 | **0.1013** | +19.74% |
| | M@20 | 0.0154 | 0.0466 | 0.0388 | 0.0409 | 0.0461 | 0.0393 | 0.0469 | _0.0622_ | 0.0481 | 0.0485 | 0.0456 | **0.0718** | +15.43% |

Table 5: Ablation study on ML1M dataset.

| Variant | N@10 | ± SE | M@10 | ± SE |
|---|---|---|---|---|
| GSPRec-NB | 0.5769 | 0.0082 | 0.5148 | 0.0093 |
| GSPRec-NL | 0.6042 | 0.0079 | 0.5229 | 0.0088 |
| GSPRec-NS | 0.6274 | 0.0039 | 0.5704 | 0.0050 |
| GSPRec-SE | 0.6416 | 0.0037 | 0.5831 | 0.0049 |
| GSPRec | **0.6431** | 0.0074 | **0.5837** | 0.0081 |

Table 6: Filtering comparison on ML1M dataset.

| Components | N@10 | ± SE | M@10 | ± SE |
|---|---|---|---|---|
| $F_{LP}$ only | 0.5769 | 0.0082 | 0.5148 | 0.0093 |
| $F_{BP}$ only | 0.6042 | 0.0079 | 0.5229 | 0.0088 |
| $F_{LP}$+$F_{BP}$ ($\phi$=0.3) | **0.6431** | 0.0074 | **0.5837** | 0.0081 |
| $F_{LP}$+$F_{BP}$ ($\phi$=0.5) | 0.6327 | 0.0076 | 0.5714 | 0.0085 |
| $F_{LP}$+$F_{BP}$ ($\phi$=0.7) | 0.6281 | 0.0078 | 0.5682 | 0.0087 |

The performance variation correlates with dataset-specific optimal parameter configurations shown at Table 3. ML100K exhibits optimal performance at lower frequency centers, Netflix at mid-range frequencies, while Beauty and ML1M favor higher frequencies. These indicate that GSPRec adapts to the underlying spectral structure of each dataset rather than imposing a fixed frequency range.

## 4.3 ABLATION ANALYSIS

Table 5 reveals a clear component hierarchy. Removing bandpass filtering causes the largest performance degradation, followed by lowpass filtering removal, while sequential diffusion removal causes the smallest drop. This hierarchy demonstrates that spectral filtering strategy contributes more substantially to performance than graph augmentation through sequential information. Table 6 confirms that bandpass filtering alone outperforms lowpass alone by 4.7%, supporting the effectiveness of mid-frequency targeting over traditional low-frequency approaches.

Parameter analysis reveals both universal architectural constants and dataset-specific adaptations. Diffusion parameters $\alpha = 0.4$ and $d = 2$ remain across all datasets, which suggests fundamental properties of sequential transition structures that transcend domain-specific characteristics. However, spectral parameters exhibit systematic variation that eigenspace dimensionality $r$ scales linearly with dataset size, while bandpass center positions cluster according to dataset characteristics, in which indicating that different datasets exhibit distinct spectral signatures in their interaction patterns.

The fusion weight $\phi$ provides interpretable insight into the balance between personalization and popularity signals. The optimal value of 0.3 on ML1M indicates that bandpass-based personalization dominates the recommendation process while lowpass popularity signals provide complementary information. This quantifies the relative importance of different spectral components and demonstrates how GSPRec achieves effective signal separation.

## 4.4 PARAMETER SENSITIVITY

Figures 4 and 5 demonstrate that optimal filter configurations depend on both center position and dataset characteristics. The interaction between center and width parameters confirms that different datasets require distinct spectral windows to capture personalized patterns while suppressing noise

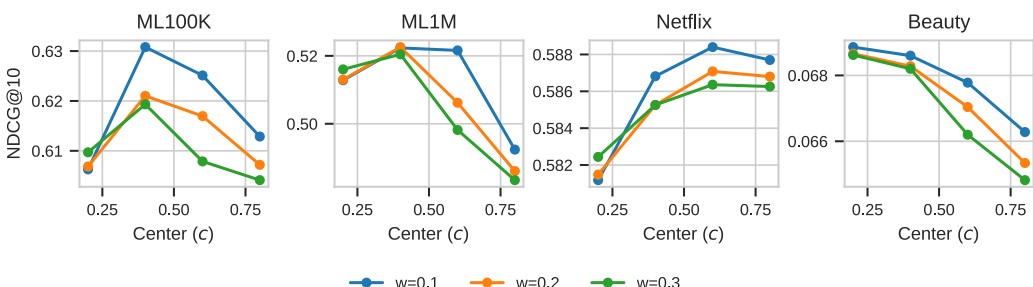

Figure 4: Effect of bandpass center position ($c$) on NDCG@10 across datasets. Each line represents a different filter bandwidth ($w$).

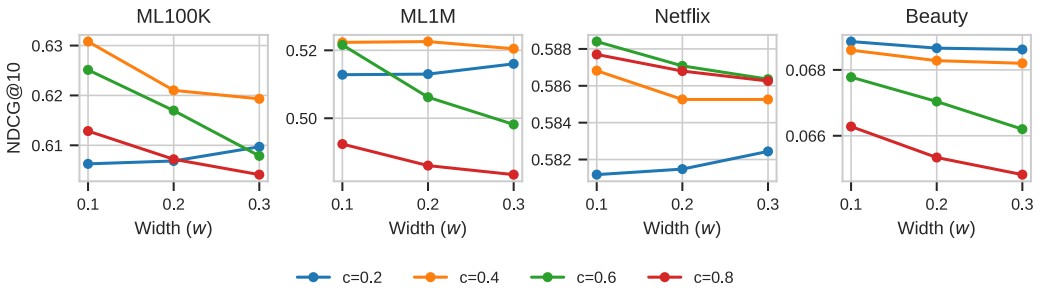

Figure 5: Effect of bandpass width ($w$) on NDCG@10 across datasets. Each line represents a different center position ($c$). Optimal width depends on center position and dataset characteristics.

amplification. This adaptive behavior explains why fixed low-pass or high-pass approaches fail to achieve consistent improvements across diverse recommendation scenarios.

Notably, the sensitivity curves exhibit relative flatness around optimal parameter values, indicating that moderate parameter deviations do not substantially degrade performance. This robustness suggests that GSPRec does not require precise hyperparameter tuning.

### 4.5 COMPUTATIONAL ANALYSIS

Table 7 shows that GSPRec requires 1.27 minutes on ML1M, comparable to other GSP methods (1.18-2.36 minutes) and substantially more efficient than GCN approaches (16-135 minutes). This positions GSPRec on the accuracy-efficiency frontier, unlike GCN methods that achieve moderate improvements at high computational cost, or GSP methods that sacrifice accuracy for speed, GSPRec provides both computational efficiency and high accuracy by targeting the most informative bands.

Table 7: Runtime comparison on ML1M dataset (minutes).

| Method | LightGCN | SimpleX | UltraGCN | GF-CF | PGSP | FaGSP | HiGSP | GSPRec |
|--------|----------|---------|----------|-------|------|-------|-------|--------|
| **Runtime** | 135.96 | 109.83 | 16.09 | 1.18 | 2.36 | 2.08 | 2.04 | **1.27** |

## 5 CONCLUSION

We presented GSPRec, a temporal-spectral collaborative filtering framework that integrates sequential transitions with dual-frequency spectral filtering. our approach consistently outperforms baselines across four datasets by targeting mid-frequency components, with bandpass filtering achieving 4.7% improvement over state-of-the-art methods.

**Limitations and Future Directions:** The symmetric construction trades directional information for spectral stability, and eigendecomposition creates computational bottlenecks addressable through polynomial filtering. This work demonstrates that restructuring graph spectra through sequential encoding enables more effective personalization than filtering static user-item graphs.

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
