# OpenReview forum: "GSPRec: Temporal-Aware Graph Spectral Filtering for Recommendation"
_ICLR.cc/2026/Conference — Submitted to ICLR 2026_

### Official Review · Reviewer_nV8G · 2025-10-29

**Soundness:** 2
**Presentation:** 3
**Contribution:** 2
**Rating:** 4
**Confidence:** 4

**Summary:**

This paper proposed a novel approach that incorporates sequential information to construct an item–item relation graph and to design a bandpass filter. By integrating sequential patterns into user–item interactions, the authors argued that the model can better capture meaningful semantic relationships among users and items. In addition, the proposed method exploited not only low-frequency components, but also mid- and high-frequency components that encode localized and task-relevant variations.

**Strengths:**

(S1) The paper proposed a new approach to modeling item–item relations by explicitly utilizing sequential information in a bidirectional manner.

(S2) The newly proposed band-pass filter was theoretically well defined, enabling the model to exploit not only low-frequency components but also mid- and high-frequency components.

**Weaknesses:**

(W1) The method increases computational complexity, while the performance gain from the multi-hop diffusion component is relatively limited.

(W2) The experimental results for hyperparameter sensitivity showed an inconsistent tendency that does not align with the implementation details.

(W3) Most importantly, due to the fact that the method leverages additional information (e.g., sequential information) that is not typically available to conventional collaborative filtering baselines, the comparison with those baselines is not entirely fair.

**Questions:**

1. The proposed method constructed item-item relation graph using sequential information in a bidirectional formulation. However, in sequential settings where order is crucial, doesn’t a bidirectional formulation break the order signal?

2. There exists a graph signal processing-based model for sequential recommendation, namely FIRE (WWW’22). However, the paper did not provide a comparison between FIRE and the proposed model. It would be valuable if the authors could include a performance comparison between the two models.

3. The design of the filter function appeared somewhat arbitrary, as it adopted an exponential-shaped form without a clear theoretical justification. In addition, when the model filtering low-frequency components, it performed eigen-decomposition for the adjacency matrix. If the eigenbasis is already available, why not directly using mid- or high-frequency components instead of designing a bandpass filter?

4. In parameter sensitivity shown in Figures 4 and 5, the proposed models with w = 0.1 appeared to achieve the best performance in most settings. Nevertheless, the models treated w as a tunable hyperparameter. In addition, Table 3 represented the filter parameters across dataset, but these reported values do not seem to match the trends observed in parameter sensitivity. Could the authors clarify why w is left as a tunable hyperparameter, and how the reported parameter choices in Table 3 were determined?

5. In Tables 5 and 6, the results included the notation “SE”, which I interpret as the standard error. However, according to my understanding, the proposed method is based on graph signal processing and does not involve a learning or training process. In that case, the model should be deterministic, and it is not clear why there would be any standard error. Could the authors clarify the source of this standard error and how it was computed?

6. In the proposed model, the band-pass filter component in Eq (8) performs normalization on the item side, while the low-pass filter in Eq (9) performs normalization that jointly considers both users and items. I agree that each choice can be justified individually. However, I am not fully convinced that it is appropriate to apply two different normalization schemes to components derived from the same adjacency structure, and then simply combine them with a weighted sum at the end.

---

> ### Author Response · Authors · 2025-11-20
>
> We thank the reviewer for their careful review and valuable feedback. We address your concerns and questions below.
>
>
> **W1: Diffusion gain**
>
> The multi-hop diffusion step adds higher-order transitions not captured by direct edges. While the gain is modest, it is consistent across datasets and introduces <5% computational overhead. This aligns with standard GSP practice, where diffusion captures higher-order locality. We will clarify this trade-off.
>
> ---
>
> **W2: Parameter consistency.**
>
> The apparent mismatch arises from comparing different types of analyses: Table 3 reports *jointly tuned* $(c,w)$, whereas Figures 4 and 5 show *1D sensitivity sweeps* where one parameter is varied while the other is fixed. Because $c$ and $w$ interact, a 1D sweep's peak does not necessarily coincide with the joint optimum. We will make this interaction more explicit in the revision.
>
> ---
>
> **W3: Comparison fairness**
> All reported baselines address the same CF task: predicting missing entries under the standard top-$N$ protocol. Our use of sequences affects only the construction of the item–item graph; the task and evaluation remain identical.
>
> Sequential models (SASRec, BERT4Rec, FIRE, etc) optimize **next-item prediction** with directional transitions and positional encodings, so they are not directly comparable as explained in section 2. We will clarify this distinction further.
>
> ---
>
> **Q1: Bidirectionality**
>
> Directionality is not retained after symmetrization, but temporal *proximity* (co-occurrence strength) is preserved through the weighted construction of $S^0$. Since our task is CF, proximity (not direction) is the relevant signal.
>
> ---
>
> **Q2: FIRE.**
>
> FIRE operates on directed transitions and is evaluated under *next-item prediction*. Our CF formulation uses undirected spectral processing, so the two settings are not directly comparable. We will add a brief note to avoid confusion.
>
> ---
>
> **Q3: Gaussian filter design**
>
> We use a Gaussian kernel because it provides smooth, stable spectral weighting and is robust to spectral perturbations [1,2]. Hard cutoffs or eigenvector selection are brittle and sensitive to noise. Gaussian windows avoid ringing artifacts and are standard in GSP. We will make this rationale explicit.
>
> ---
>
> **Q4: $w$**
>
> While small $w$ may perform well in isolation, the best \emph{joint} $(c,w)$ depends on the dataset’s spectral structure. The values in Table 3 come from a two-parameter grid search. We will clarify that $w$ is tuned jointly with $c$.
>
> ---
>
> **Q5: Meaning of SE**
>
> The $\pm$ SE values denote *standard error across users*. The model is deterministic, but user-level metrics vary across the test set. We report $\mathrm{SE}=\sigma/\sqrt{N}$ to reflect population variability. To avoid confusion, we will rename the Sequential Edges ablation to $S^{0}$.
>
> ---
>
>
> **Q6: Normalization choices**
>
> The bandpass and lowpass components use the same augmented graph and share the same eigenbasis. They differ in how the signal is normalized before/after spectral filtering: the *bandpass component* uses item-side normalization, which reflects its origin in the item-item sequential subgraph, while the *lowpass component* uses bipartite normalization, which is appropriate for aggregating user-item information. Both filtered outputs reside in the item domain and are combined after filtering.
>
> This separation is standard in multi-operator GSP, where each component uses the normalization appropriate for its graph geometry before combining filtered signals in the vertex domain.
>
> ---
>
>
>
> **References**
>
> [1] David K Hammond, Pierre Vandergheynst, and Remi Gribonval. Wavelets on graphs via spectral ´graph theory. Applied and Computational Harmonic Analysis, 30(2):129–150, 2011.
>
> [2] David I Shuman, Sunil K Narang, Pascal Frossard, Antonio Ortega, and Pierre Vandergheynst. The emerging field of signal processing on graphs: Extending high-dimensional data analysis to networks and other irregular domains. IEEE signal processing magazine, 30(3):83–98, 2013.

---

> > ### Comment · Reviewer_nV8G · 2025-11-25
> > **Further feedback**
> >
> > Thank you for your responses. I have two remaining concerns below.
> >
> > 1. Comparison fairness:
> >
> >  The baseline models rely solely on the user–item interaction matrix (often denoted R), whereas the proposed method
> > appears to exploit additional information beyond these raw interactions. Intuitively, a model that is allowed to use richer input information would be expected to achieve higher recommendation accuracy than models restricted to R alone.
> >
> > 2. Bidirectionality:
> >
> > I still believe that once we decide to use sequential information, the actual "order" in which items are consumed becomes important. To give a simple example using food: in most cases, people eat in the order of appetizer->main course->dessert. Yet, under the methodology proposed in this paper, it seems that a sequence like dessert->main course->appetizer would still  be treated as "sequential information", which is somewhat counterintuitive from this perspective.

---

> > > ### Author Response · Authors · 2025-11-30
> > >
> > > Thank you for the follow-up. The comments highlight aspects of the presentation that we can clarify further, and we will revise those sections accordingly.
> > >
> > > ---
> > >
> > > **On comparison fairness:**  Our ablations separate the effects of sequential augmentation from the effects of our spectral architecture. Even when we remove sequential edges, the dual-frequency filtering alone outperforms the strongest CF baselines, which shows that the architectural contribution is not dependent on sequence information. Sequential augmentation adds an additional improvement while the main performance gain comes from the filtering mechanism itself.
> > >
> > > More broadly, temporal information has been incorporated into CF models in various ways. For example, through timestamp features as in Factorization Machines [1], temporal dynamics [2,3], or session-level proximity patterns [4]. Our use of sequences is similar in spirit: we use them solely to construct an item-item graph prior to spectral filtering, without altering the matrix-completion objective or evaluation protocol. We will revise the paper to more clearly articulate this distinction and explain how our use of temporal proximity fits into existing CF practice.
> > >
> > >
> > > ---
> > >
> > > **On bidirectionality:** Symmetrization is a standard requirement in spectral methods when a real, orthogonal eigenbasis is needed [5,6]. In our setting, the symmetrized graph encodes temporal proximity (items that co-occur within short windows), which is the form of temporal information relevant for CF. Unlike next-item prediction (which requires directional transitions because the task is causal), CF relies on affinity patterns between items, where order is often noisy or non-informative (e.g., movies rated in arbitrary order, purchases interleaved across sessions).
> > >
> > > We will revise Section 3.2 to state explicitly that our goal is to incorporate temporal locality rather than directional causality, and that sequences are used only to enrich the item-item graph before spectral filtering. This clarification aligns the paper more closely with established practice in CF and spectral methods.
> > >
> > > ---
> > >
> > > **References:**
> > >
> > > [1] Steffen Rendle. Factorization machines. In ICDM, 2010.
> > >
> > > [2] Yehuda Koren. Collaborative filtering with temporal dynamics. In KDD, 2009.
> > >
> > > [3] Liang Xiong, Xi Chen, Tzu-Kuo Huang, Jeff Schneider, and Jaime G. Carbonell. Temporal collaborative filtering with Bayesian probabilistic tensor factorization. In SDM, 2010.
> > >
> > > [4] Balázs Hidasi, Alexandros Karatzoglou, Linas Baltrunas, and Domonkos Tikk. Session-based recommendations with recurrent neural networks. In ICLR, 2016.
> > >
> > > [5] Ulrike Von Luxburg. A tutorial on spectral clustering. Statistics and Computing, 17(4):395-416, 2007.
> > >
> > > [6] Venu Satuluri and Srinivasan Parthasarathy. Symmetrizations for clustering directed graphs. In EDBT, 2011.
> > >
> > > [7] Wang-Cheng Kang and Julian McAuley. Self-attentive sequential recommendation. In ICDM, 2018.
> > >
> > > [8] Fei Sun, Jun Liu, Jian Wu, Changhua Pei, Xiao Lin, Wenwu Ou, and Peng Jiang. BERT4Rec: Sequential recommendation with bidirectional encoder representations from transformer. In CIKM, 2019.

---

### Official Review · Reviewer_oPrG · 2025-10-31

**Soundness:** 2
**Presentation:** 3
**Contribution:** 2
**Rating:** 4
**Confidence:** 4

**Summary:**

This paper proposes GSPRec, a temporal-aware graph spectral filtering approach for collaborative filtering. The method addresses limitations of existing graph-based methods that rely primarily on user-item interactions and low-pass filtering. The key contributions are: (1) a sequential graph construction approach that integrates temporal transition patterns from user sequences through symmetrization and multi-hop diffusion, capturing both direct and indirect item relationships; (2) a novel bandpass filtering design that targets mid-frequency spectral components to extract community-level personalization patterns, distinguishing it from prior work using low-pass or high-pass filtering; (3) a dual-filter architecture combining Gaussian bandpass and lowpass filters to balance personalized recommendations with global popularity. Experiments on four datasets demonstrate consistent performance improvements over baselines, with ablation studies showing bandpass filtering as the primary contributor. The work offers a frequency-aware perspective for analyzing collaborative signals in recommendation systems.

**Strengths:**

1. The paper introduces bandpass filtering to recommendation systems, explicitly targeting mid-frequency components—a distinct approach from prior GSP methods that focus on low-pass or high-pass filtering. The dual-filter architecture combining bandpass and lowpass filters represents a new design for balancing personalization and popularity in spectral collaborative filtering.

2. The paper offers a new lens for understanding collaborative signals through spectral analysis, proposing that different frequency components may capture different aspects of user preferences (e.g., low-frequency for global popularity, mid-frequency for community patterns). While the universality of this hypothesis requires further investigation (as noted in Weaknesses), this frequency-aware perspective provides a valuable framework for analyzing recommendation signals.

3. Experimental results demonstrate the method's effectiveness across multiple datasets with varying characteristics. The approach shows consistent improvements over baseline methods, and ablation analysis indicates that bandpass filtering contributes substantially to overall performance. The results suggest that the proposed frequency-aware filtering approach offers practical benefits for recommendation tasks.

**Weaknesses:**

1.  Unclear Utilization of Temporal Information
	The paper claims to be "Temporal-Aware" throughout, but the utilization of temporal order is not clearly demonstrated. Lines 196-198 symmetrize the sequential transition matrix (S⁰[i,j] = 1 if i→j OR j→i exists), and Equation (3) produces a symmetric matrix S̃ where S̃[a,b] = S̃[b,a]. While the authors note this symmetrization is required for spectral analysis (lines 183-185), it remains unclear how the method distinguishes "a→b" from "b→a" given that undirected graphs with symmetric adjacency do not naturally encode directionality. The method may primarily capture sequential co-occurrence rather than temporal order. The authors should clarify how directional information is preserved or acknowledge this as a limitation.

2.Parameter Inconsistencies with Sensitivity Analysis
	Figure 4 reveals apparent contradictions with Table 3 for the Beauty dataset. The sensitivity curves suggest better performance at lower c values (around c=0.25 with w=0.1), while Table 3 reports c=0.8 and w=0.3 as optimal. Additionally, the monotonic decrease in Figure 4 appears inconsistent with the paper's hypothesis that mid-frequency (c≈0.5) captures optimal personalization (lines 294-299). This raises questions about: (1) which parameters were used for Table 4 results, (2) whether the mid-frequency hypothesis applies to all datasets, and (3) whether bandpass filtering is necessary for Beauty. Beauty-specific ablation experiments (analogous to Table 5) would help clarify these points.

3. Missing Comparisons with Sequential Baselines
	Table 1 lists BERT4Rec and SASRec as relevant methods, but Table 4 does not include experimental comparisons with them. Given that these methods preserve temporal order through positional encodings and attention mechanisms, they would serve as appropriate baselines for evaluating "Temporal-Aware" claims. Comparisons with these sequential recommendation approaches would help assess the trade-offs between the proposed graph-based approach and methods that explicitly maintain temporal information.

**Questions:**

1.How is directional information from sequences preserved after symmetrization, and should the "Temporal-Aware" claim be reconsidered?
2.What explains the parameter discrepancy between Table 3 (c=0.8, w=0.3) and Figure 4 (suggesting c≈0.25, w=0.1) for Beauty? Which parameters were used for Table 4?
3.Why were BERT4Rec and SASRec (listed in Table 1) not included in experimental comparisons?
4.Can the authors provide Beauty-specific ablation experiments and clarify the definitions of ablation variants (GSPRec-NB/NL/NS/SE)?

---

> ### Author Response · Authors · 2025-11-20
>
> We thank the reviewer for their detailed review and insightful questions. We appreciate your feedback and address your points below.
>
> **1. Directionality vs. temporal-aware signal**
>
> Symmetrization does remove directionality as noted in Section 3.2. This is necessary for a stable spectral operator and is standard in GSP-based CF models. The temporal information we retain is *proximity*: items that appear close together in user histories receive larger weights in $S^{0}$ via the $\alpha^{k-1}$ window, so short-range co-occurrence is preserved even though the graph is undirected.
>
> Since our task is collaborative filtering (matrix completion), temporal proximity (not causal order) is the relevant notion of similarity. Modeling direction corresponds to next-item prediction and requires a different framework. We will make this distinction clearer.
>
> ---
>
> **2. Parameter behavior on Beauty**
>
> The discrepancy arises from comparing two diagnostics. Table 3 reports the *jointly tuned* $(c,w)$ from a 2D grid search, whereas Figures 4 and 5 show *1D sensitivity sweeps* that vary a single parameter while holding the other fixed. Because $c$ and $w$ interact, peaks from 1D sweeps are not expected to match the joint optimum.
>
> Beauty’s extreme sparsity (0.07%) shifts its effective spectral region, making low-$c$ settings appear stronger in isolation. The Beauty component hierarchy matches Table 5. We will make this explicit in the revision.
>
> ---
>
> **3. Sequential baselines**
>
> Sequences in GSPRec are used only to construct the item-item graph as described in section 2. The prediction task remains collaborative filtering (matrix completion). Sequential models such as SASRec and BERT4Rec optimize a different objective (next-item prediction) and follow a different evaluation protocol. Since the task definition differs, they are not directly comparable baselines. We will clarify this distinction.
>
> ---
>
> **4. Definitions of ablation variants**
>
> Thank you for the note. NB = no bandpass; NL = no lowpass; NS = no sequential augmentation; SE = sequential edges only ($S^0$ without diffusion).
> We will make these definitions explicit in the main text for clarity.

---

### Official Review · Reviewer_3fBT · 2025-11-04

**Soundness:** 3
**Presentation:** 2
**Contribution:** 3
**Rating:** 4
**Confidence:** 2

**Summary:**

This paper proposes GSPRec, a graph-spectral recommendation model that first turns user sequences into a symmetric item-item graph via multi-hop diffusion and then applies a dual spectral filter, which is a Gaussian bandpass to mine mid-frequency, user-specific patterns and a lowpass to keep global popularity, before fusing both signals to predict the next interaction.

**Strengths:**

1. By transforming raw click sequences into a symmetric graph before spectral filtering, GSPRec preserves temporal cues while maintaining the stability of eigendecomposition.

2. The entire pipeline requires only matrix decomposition and lightweight filters, demonstrating high efficiency as shown in the experiments.

3. The conducted experiments demonstrate the superior performance of the proposed method.

**Weaknesses:**

1. Throughout the paper the mid-frequency Gaussian bandpass is said to capture ‘user-specific sequential patterns’. Is there any case study to demonstrate it?

2. The experiments only report aggregate top-k accuracy, but emphasizing mid-frequency components may systematically boost long-tail items and thus change the exposure distribution. Could the authors include fairness-aware metrics in the experiments?

**Questions:**

See Weakness.

---

> ### Author Response · Authors · 2025-11-20
>
> We thank the reviewer for their thoughtful review and constructive feedback. We address your comments below.
>
> **W1: Case study for mid-frequency effects**
>
> Thank you for this suggestion. Our ablations isolate the contribution of mid-frequency components: the bandpass-only variant consistently outperforms the lowpass-only variant (Table 6), and the largest relative gain occurs on the sparsest dataset (Beauty, 0.07% density). This aligns with the GSP view that:
> - low frequencies capture global popularity,
> - high frequencies largely reflect noise or sharp transitions, and
> - mid frequencies localize on item communities and thus encode more personalized or niche structure. The current results already support this interpretation. For completeness, we will include a brief illustrative example in the appendix to clarify the mid-frequency behavior more explicitly.
>
> ---
>
> **W2: Fairness metrics**
>
> We appreciate the question. Our evaluation focuses on top-$k$ accuracy, and the paper does not make fairness or exposure claims. While emphasizing mid-frequency components may affect long-tail exposure, assessing fairness would require a dedicated experimental design and is beyond the scope of the present work. We will clarify this scope to avoid unintended interpretations.

---

### Meta-Review · Area_Chair_rKzu · 2026-01-06

**Summary:**

This paper proposed a graph spectral model that integrates temporal transitions through sequentially-informed graph construction and applies frequency-aware filtering in the spectral domain named GSPPec for recommendation. 3 Reviewers provided detailed comments on the paper, and all of the reviewers gave score 4. After reading the reviews and the author rebuttal, I think many concerns of the reviewers are not well addressed and remain. Therefore, I recommend to reject the paper.

**Reviewer Concerns:**

Outstanding:
Reviewer 3fBT: Case study for mid-frequency Gaussian bandpass, include fairness-aware metrics in the experiments
Reviewer oPrG: Sequential baselines, Unclear Utilization of Temporal Information
Reviewer nV8G: Comparison fairness and Bidirectionality, more baselines

**Reviewer Scores:**

I think it is very difficult for the reviewers to change their scores after the rebuttal, as most of their concerns are not well addressed.

---

### Decision · Program_Chairs · 2026-01-26

Reject